# Immuno-Modulatory Effects of Dexamethasone in Severe COVID-19—A Swedish Cohort Study

**DOI:** 10.3390/biomedicines11010164

**Published:** 2023-01-09

**Authors:** Sana Asif, Robert Frithiof, Anders Larsson, Stephanie Franzén, Sara Bülow Anderberg, Bjarne Kristensen, Michael Hultström, Miklos Lipcsey

**Affiliations:** 1Anaesthesiology and Intensive Care Medicine, Department of Surgical Sciences, Uppsala University, 751 85 Uppsala, Sweden; 2Department of Medical Sciences, Uppsala University, 751 85 Uppsala, Sweden; 3Thermo Fisher Scientific, DK-84 3450 Alleröd, Denmark; 4Unit for Integrative Physiology, Department of Medical Cell Biology, Uppsala University, 751 85 Uppsala, Sweden; 5Hedenstierna Laboratory, Department of Surgical Sciences, Uppsala University, 751 85 Uppsala, Sweden

**Keywords:** COVID-19, dexamethasone, treatment timing, intensive care, hypoxia, cytokines

## Abstract

Dexamethasone (Dex) has been shown to decrease mortality in severe coronavirus disease 2019 (COVID-19), but the mechanism is not fully elucidated. We aimed to investigate the physiological and immunological effects associated with Dex administration in patients admitted to intensive care with severe COVID-19. A total of 216 adult COVID-19 patients were included—102 (47%) received Dex, 6 mg/day for 10 days, and 114 (53%) did not. Standard laboratory parameters, plasma expression of cytokines, endothelial markers, immunoglobulin (Ig) IgA, IgM, and IgG against SARS-CoV-2 were analyzed post-admission to intensive care. Patients treated with Dex had higher blood glucose but lower blood lactate, plasma cortisol, IgA, IgM, IgG, D-dimer, cytokines, syndecan-1, and E-selectin and received less organ support than those who did not receive Dex (Without-Dex). There was an association between Dex treatment and IL-17A, macrophage inflammatory protein 1 alpha, syndecan-1 as well as E-selectin in predicting 30-day mortality. Among a subgroup of patients who received Dex early, within 14 days of COVID-19 debut, the adjusted mortality risk was 0.4 (95% CI 0.2–0.8), i.e., 40% compared with Without-Dex. Dex administration in a cohort of critically ill COVID-19 patients resulted in altered immunological and physiologic responses, some of which were associated with mortality.

## 1. Introduction

Corticosteroid treatment has been shown to reduce morbidity and mortality among patients hospitalized with severe coronavirus disease 2019 (COVID-19) through steroid-mediated downregulation of local and systemic inflammatory response and by restoring tissue homeostasis [1,2,3]. The Recovery trial provided evidence that in patients hospitalized with severe COVID-19, treatment with dexamethasone (Dex), 6 mg/day up to 10 days or until discharge, reduced mortality [4]. These data are supported by multicenter studies from France and Brazil [5,6]. Similar protective effects were reported in the REMAP-CAP trial which included 403 patients treated with corticosteroids and admitted to intensive care due to severe COVID-19 [7]. The rationale for corticosteroid use in COVID-19 treatment was partially based on previous studies of acute respiratory distress syndrome (ARDS), sepsis, or acute fibrinous and organizing pneumonia [8]. However, the beneficial effects reported from these studies are largely controversial and mixed conclusions have been drawn. For COVID-19 patients, reports from Wang et al. [9,10] demonstrated no beneficial effects of corticosteroids on mortality, but delayed viral clearance and a higher frequency of opportunistic infections. Interpretation of the clinical effects of broad anti-inflammatory drugs such as corticosteroids necessitates an understanding of the key mechanisms of action in the specific setting where they are used. 

The observation that corticosteroid treatment may be beneficial in severely ill patients could be explained by the multifaceted effects of steroids that target different pathophysiological components of COVID-19 [11,12]. Corticosteroids exert anti-inflammatory effects by modulating both innate and adaptive immune responses through suppressing production of many inflammatory mediators as well as their downstream signaling pathways [13,14]. They also enhance the clearance of foreign antigens, toxins, and dead cells from inflamed sites by promoting opsonization and stimulating macrophage phagocytosis [15].

COVID-19 has been shown to cause endothelial dysfunction through direct viral endothelial injury, uncontrolled immune and inflammatory responses, coagulation imbalances, disruption of vascular hemostasis, and crosstalk between these systems, resulting in a viscous cycle exacerbating the disease process, ultimately resulting in organ failure and death [16]. Dex treatment has been shown to stabilize endothelial function by inhibiting inflammatory glycocalyx shedding [17]. Furthermore, low-dose corticosteroids are beneficial in restoring the balance between the hypothalamic-pituitary axis and cortisol secretion [18].

Corticosteroid treatment thus seems to be a double-edged sword and careful risk-benefit assessments must be made before it is initiated. We hypothesized that the mechanisms which mediated the effects of Dex treatment on the risk of death in COVID-19 could be identified in a natural experiment, as corticosteroids were not used prior to the Recovery study [4], but were given to all patients thereafter. 

Consequently, we re-analyzed the prospective PronMed cohort as a before-after intervention study to investigate possible mediators of the physiological and immunological effects of Dex on short-term mortality in patients with severe COVID-19 admitted to an intensive care unit (ICU). 

## 2. Methods

This is a sub-study of the single-center, prospective observational investigation PronMed study, approved by the Swedish National Ethical Review Agency, Dnr 2017-043 (with amendments 2019-00169, 2020-01623, 2020-02719, 2020-05730, 2021-01469) and Dnr 2022-00526-01. Informed consent was obtained either from the patient or from next-of-kin if the patient was unable to receive information due to their clinical status. The Declaration of Helsinki and its subsequent revisions were followed. The protocol of the study was registered a priori (Clinical Trials ID: NCT04316884). STROBE guidelines were applied in reporting.

### 2.1. Patients

Adult (18 years or older) patients with ongoing COVID-19 who were admitted to the ICU between 14 March 2020 and 11 January 2021 were included in this study. COVID-19 was diagnosed through a positive reverse transcriptase polymerase chain reaction for severe acute respiratory syndrome coronavirus 2 (SARS-CoV-2) on nasopharyngeal swabs. 

### 2.2. Exposure

Patients were not treated with corticosteroids during the initial phase of the COVID-19 pandemic. After the publication of the dexamethasone arm of the Recovery study [4], all patients received Dex. Thus, we designed the present analysis as a before-after intervention study; all patients admitted to the ICU after 22 June 2020 received Dex treatment. Based on time of inclusion, the patients included in this study were divided into two groups: those who received Dex (With-Dex) and those who did not (Without-Dex). Dex treatment was generally initiated when patients first needed supplemental oxygen, i.e., before arrival to the ICU.

### 2.3. Primary Outcome

The primary outcome was interaction effect between biomarkers and Dex administration on 30-day mortality in patients admitted to the ICU as critically ill with COVID-19. A secondary aim was to identify if the timing of Dex administration was associated with 30-day mortality in critically ill COVID-19 patients.

### 2.4. Baseline Characteristics

Baseline parameters such as age, sex, and body mass index (BMI) were recorded on admission. Information on comorbidities such as lung disease, hypertension, ischemic heart disease, diabetes mellitus, and chronic kidney disease (CKD) was extracted from medical records. CKD was defined as a baseline estimated glomerular filtration rate of <60 mL/min/1.73 m^2^, corresponding to CKD stage 3 or worse [19]. COVID-19 day on admission to the ICU refers to the number of days from the reported first symptom to ICU admission. Information on COVID-19 day was extracted from medical records.

### 2.5. Clinical Variables and Biochemistry

Data on illness severity (simplified acute physiology score 3, SAPS3) [20], continuous renal replacement therapy (CRRT), use of vasopressors, mechanical ventilation, mortality at 30 days, secondary infections, and length of ICU stay were collected over time, as reported in the results. Lung function was assessed based on the ratio of arterial oxygen partial pressure (PaO_2_) in mmHg to the fraction of inspired oxygen (FiO_2_) and was recorded on admission to ICU [21]. Blood cell counts (CBC), high-sensitivity C-reactive protein (hsCRP), procalcitonin, plasma sodium, potassium, glucose, creatinine, cystatin-C, lactate, fibrin D-dimer, and ferritin analyses were performed daily during ICU stay, in the hospital’s central laboratory. CBC was analyzed on a Sysmex XN™ instrument (Sysmex, Kobe, Japan) whereas hsCRP was quantified on an Architect ci16200 (Abbott Laboratories, Abbott Park, IL, USA).

Plasma E-selectin was analyzed using an enzyme-linked immunosorbent assay (ELISA) from R&D Systems (DY724, Abingdon, UK). The limit of detection for the assay was 90 pg/mL.

Plasma syndecan-1 was analyzed using an ELISA from R&D Systems (DY2780). The limit of detection was 120 pg/mL.

Plasma cortisol was measured using a Parameter ELISA from R&D Systems (KGE008B). The limit of detection was 0.071 ng/mL.

### 2.6. Cytokine Assay

Plasma samples were analyzed for 27 biomarkers with Bioplex assay using a Luminex MagPix instrument (Bio-Rad laboratories AB, Sundbyberg, Sweden). The following cytokines were analyzed on day seven after ICU admission: interleukin (IL)-1β, IL-2, IL-4 to IL-10, IL-12, IL-13, IL-15, IL-17A, interferon gamma (IFN-γ), IFN-γ-induced protein 10 (IP-10), monocyte chemotactic protein 1, macrophage inflammatory protein 1 alpha (MIP-1α), macrophage inflammatory protein 1 beta (MIP-1β), tumor necrosis factor (TNF), IL-1 receptor antagonist, normal T cell expressed and secreted, platelet-derived growth factor BB, basic fibroblast growth factor, granulocyte-macrophage colony-stimulating factor, vascular endothelial growth factor, and eotaxin. Cytokine expression is reported as ng/mL, the lower limits of detection for the cytokines analyzed were: IL-1β (0.15 ng/mL), IL-2 (0.05 ng/mL), IL-4 (0.13 ng/mL), IL-6 (0.02 ng/mL), IL-8 (0.06 ng/mL), IL-10 (0.02 ng/mL), IL-17A (0.002 ng/mL), IFN-γ (0.14 ng/mL), MIP-1α (0.2 ng/mL). For some patients, cytokine assays could not be performed due to limited plasma sample volume. 

### 2.7. COVID-19 Antibodies

Plasma samples were collected on days 15–20 after ICU admission, and concentrations of IgA, IgG and IgM antibodies were quantified using fluoro-enzyme immunoassay (Phadia AB, Uppsala, Sweden). The lower limits of detection were 5 and 20 µg/L for IgA and IgM, respectively, and 10 U/L for IgG.

### 2.8. Statistics

Continuous variables are presented as median (interquartile range) or mean ± standard deviation, as appropriate, while categorical variables are presented as *n* (%). Fischer’s exact test or the Mann–Whitney U-test was used for group comparisons. The study size was determined by the number of patients available with the ancestral virus variant, with or without Dex treatment.

To identify possible mediators of Dex treatment, logistic regression models for 30-day mortality, adjusted for possible confounders (SAPS3, comorbidities and Dex treatment), were used with variables including biomarkers and outcomes that differed in crude With-Dex vs. Without-Dex comparisons. Comorbidities is a composite variable and represents medical history of hypertension, ischemic heart disease, and/or diabetes mellitus. A significant effect between 30-day mortality and a variable suggests an association between death and the variable, while a significant interaction effect between Dex and a variable on 30-day mortality suggests that the association between death and a variable is, statistically, dependent on Dex effect. 

To investigate the possible effect of the timing of Dex administration and 30-day mortality, a sub-analysis was conducted. For this, patients in the With-Dex group were subgrouped into Early-Dex, patients who received Dex earlier in their disease course (COVID-19 day < 14 days), and Late-Dex, patients who received Dex treatment late (COVID-19 day ≥ 14 days). The 14-day cut-off was chosen arbitrarily to group patients with early ARDS and those with fully developed ARDS.

Data plotting and statistical analyses were performed using Prism version 6 for Macintosh software (Graphpad, San Diego, CA, USA). Comparisons between groups were considered to be significant when *p* < 0.05. Given the exploratory design of the study, no correction was made for multiple comparisons. 

## 3. Results

### 3.1. Study Population

A total of 216 patients was recruited to this cohort (inclusion rate 80%), 102 (47%) With-Dex and 114 (53%) Without-Dex. Baseline characteristics are shown in Table 1. With-Dex patients were on average older and sicker than Without-Dex patients. No differences in sex or BMI were found. A higher baseline prevalence of hypertension and cardiovascular disease was found among With-Dex patients than among Without-Dex patients.

### 3.2. Organ Support and Crude Outcomes

Fewer patients in the With-Dex group were on mechanical ventilation (Table 2). Similarly, fewer patients were treated with vasopressors or CRRT. No difference was found in incidence of secondary infections. Length of stay in the ICU was shorter in With-Dex patients. No difference in crude 30-day mortality was found.

### 3.3. Laboratory and Biochemical Parameters

The laboratory and biochemical data are presented in Table 2. Patients in the With-Dex group had lower plasma levels of hsCRP, but higher levels of procalcitonin and cystatin-C. Blood glucose measured at day one after ICU admission was higher in With-Dex patients. Plasma samples collected at day seven after ICU admission showed lower levels of cortisol, ferritin, and D-dimer in With-Dex patients.

### 3.4. Cytokines and Endothelial Markers

Plasma levels of cytokines (IL-1β, IL-2, IL-6, IL-8, IL-10, IL-17A, IFN-γ, MIP-1α) were lower in the With-Dex group than in the Without-Dex group (Figure 1). Similarly, plasma concentrations of endothelial and glycocalyx markers E-selectin and syndecan-1 were lower in the With-Dex group than in the Without-Dex group (Figure 1).

### 3.5. COVID-19 Antibodies

Reduced plasma concentrations of IgA, IgM, and IgG antibodies were observed in With-Dex patients (Table 2).

### 3.6. Logistic Regression Models

In an analysis adjusted for confounders (Dex treatment, comorbidities, and SAPS3), D-dimer and syndecan-1 were independent predictors of increased mortality, whereas blood lactate was an independent predictor of decreased mortality (Figure 2A).

There was interaction between Dex treatment and IL-17A, MIP-1α, syndecan-1, and E-selectin, respectively on mortality (Figure 2B).

Dex treatment was not a predictor of 30-day mortality in the whole cohort (OR (95% CI): 0.7 (0.4–1.4), Table 3 (A)), even after adjusting for confounders (comorbidities, SAPS3). However, in With-Dex patients, Early-Dex administration was an independent predictor of survival (Table 3 (B)), while this association was not seen in Late-Dex.

## 4. Discussion

This before-after study showed that Dex treatment in patients with severe COVID-19 was associated with widespread effects on the innate and adaptive immune, endocrine, metabolic, and coagulation systems, as well as on endothelial glycocalyx, vascular permeability, and organ failure. Disruption in the aforementioned homeostatic systems was associated with mortality. The effects on biomarkers of metabolism, coagulation, and endothelial glycocalyx and their associations with mortality persisted even after adjusting for confounding factors such as severity of illness, comorbidities, and Dex, suggesting that these effects were independent of Dex treatment. In patients receiving Dex treatment, we found associations between Dex and biomarkers of innate and adaptive immune response as well as endothelial injury which subsequently affected mortality, suggesting that these pathways might be mediators of Dex’s effects on mortality. Lastly, Early-Dex-patients may have had a lower risk of mortality than was seen for Late-Dex patients.

Dex treatment aids in restoring vascular homeostasis after injury and infection [22]. The endothelial glycocalyx has a key role in regulating vascular permeability and cell adhesion and possesses anti-thrombotic properties [23,24,25]. Our results showed lower plasma levels of E-selectin, an adhesion molecule normally found in the endothelial luminal membrane, and syndecan-1, a marker of glycocalyx shedding, in With-Dex patients than in Without-Dex patients.

Our results also showed that With-Dex patients had lower plasma levels of pro-inflammatory cytokines such as IL-17A and MIP-1α. This might create the beneficial effects on mortality by inhibiting hyperinflammation. IL-17A is a multifunctional cytokine produced by T lymphocytes [26,27]. Dysregulation in the Th17 cells and production of IL-17A in the endothelium promotes expression of many downstream pro-inflammatory cytokines, such as IL-8, TNF, and IL-6, leading to tissue necrosis [28]. Moreover, Th17 cells are important in counteracting viral translocation through mucosal barriers in other viral diseases, which in theory could also be of importance in COVID-19 [29]. Likewise, MIP-1α secreted by mononuclear phagocytes exerts chemotactic and stimulatory effects on polymorphonuclear cells [30], suggesting that the innate immunity is modulated in a beneficial way by Dex treatment.

We reported lower levels of anti-SARS-CoV-2 IgM levels in With-Dex than Without-Dex patients. Corticosteroid treatment suppresses cellular immunity (Th1 cells) and promotes humoral immunity (Th2 cells) [31]. Studies on patients hospitalized with COVID-19 showed no delay or attenuation in the antibody response [32,33]. In our study, IgM but not IgA or IgG levels were lower in the With-Dex group. Given that patients in this group were admitted to the ICU on average two days later than Without-Dex patients, the difference in the early antibody response could be in part be explained by the duration of COVID-19. Moreover, lower antibody titers in the With-Dex group were not associated with mortality.

Our study demonstrated that Dex treatment reduces plasma cortisol levels, as might be expected. Corticosteroid treatment affects regulation of cortisol secretion at the hypothalamic-pituitary axis. The effects of this in critically ill COVID-19 patients is uncertain. Elevations of plasma cortisol levels cause dysregulated innate immune response to viral infections and has been shown to be linked to adverse outcomes in severe COVID-19 [34].

Dex treatment preserves vascular function and thus protects against illness severity [35,36]. We saw less respiratory, circulatory, and renal organ support as well as shorter ICU stays in With-Dex patients than in Without-Dex patients. The lower frequency of mechanical ventilation and vasopressor use could partly be explained by an increased use of non-invasive ventilation as the pandemic continued. Nevertheless, our results are in line with those of the Recovery study [4], where Dex administration was associated with lower risk of mechanical ventilation and 30-day mortality in hospitalized patients.

However, many questions remain in regard to which corticosteroid drug to use, its dosing, timing, and patient selection [37]. Dex treatment does not affect mortality. However, we showed a beneficial role of Dex administration in the Early-Dex group in comparison to Late-Dex.

Adverse effects of Dex treatment were also present in our study. Corticosteroid treatment is known to cause hyperglycemia [38,39], and higher blood glucose levels were indeed observed in With-Dex patients. Corticosteroid treatment also increased cystatin-C levels, as we and others have shown before, although the impact of this on outcome is uncertain [40,41]. The changes in cystatin-C did not correlate with changes in plasma creatinine, implying that changes in renal function do not explain this phenomenon. Still, this can be of relevance, since higher cystatin-C levels without changes in glomerular filtration rate could potentially affect decisions on drug dosing, e.g., lower doses of antimicrobial agents, leading to undertreatment. Corticosteroid treatment is associated with secondary infections [42]. Forty-four percent of With-Dex patients developed hospital-acquired infections, though this figure was not higher than that in Without-Dex patients and was similar to previously reported frequencies [38,43].

### 4.1. Strengths and Limitations

This study investigated the possible physiological and immunological mechanisms by which Dex treatment affects short-term mortality in severe COVID-19. It is the first of its kind and may be useful in designing future studies that further elucidate the biological effects of Dex in COVID-19 with ARDS. Strengths include the pragmatic characteristics of a natural experiment, making the risk of performance bias lower than in other study designs.

A further key asset of this study is the granular clinical, physiological, and immunological data on all patients included in this study. A majority of COVID-19 patients admitted to the ICU during the study were included, without patient selection, and no patients were lost to follow-up.

The study also has limitations. It is an observational study with inherent limitations in showing causality. Still, broad panel studies are important for hypothesis generation. Although many factors were constant during the study period. there were some changes in ventilation strategies, thrombosis prophylaxis, and adjunct COVID-19 treatments during the study that could potentially affect our findings.

This study is also a single-center study, where the majority of the patients were Swedish men. Though this could limit the external validity of the study, the cohort represents the demographic of COVID-19 patients typically seen in Sweden [44]. Lastly, the study has a relatively small cohort size, limiting its power. However, the window of opportunity to perform this study in unvaccinated patients with or without steroid treatment has passed and recruitment of new patients that fit the study design is no longer possible. 

### 4.2. Clinical Implications

Our study identified pathways that could be key mechanisms for improving outcome in COVID-19. If the roles of these pathways are confirmed in future studies, more specific treatments than corticosteroids, with less broad effects, can potentially improve the risk-benefit ratio of anti-inflammatory treatment for COVID-19. Furthermore, the present study illustrates the importance of appropriate timing of Dex treatment.

## 5. Conclusions

In a cohort of severely ill COVID-19 patients, we have shown that Dex treatment is associated with a variety of clinical, physiological, and immunological effects that may contribute to both the beneficial effect of treatment and its side effects. Changes in the innate and adaptive immune response as well as endothelial function and ongoing Dex treatment were associated with effects on mortality. Lastly, Early-Dex administration was associated with decreased risk of mortality in patients admitted to an ICU in Sweden.

## Figures and Tables

**Figure 1 biomedicines-11-00164-f001:**
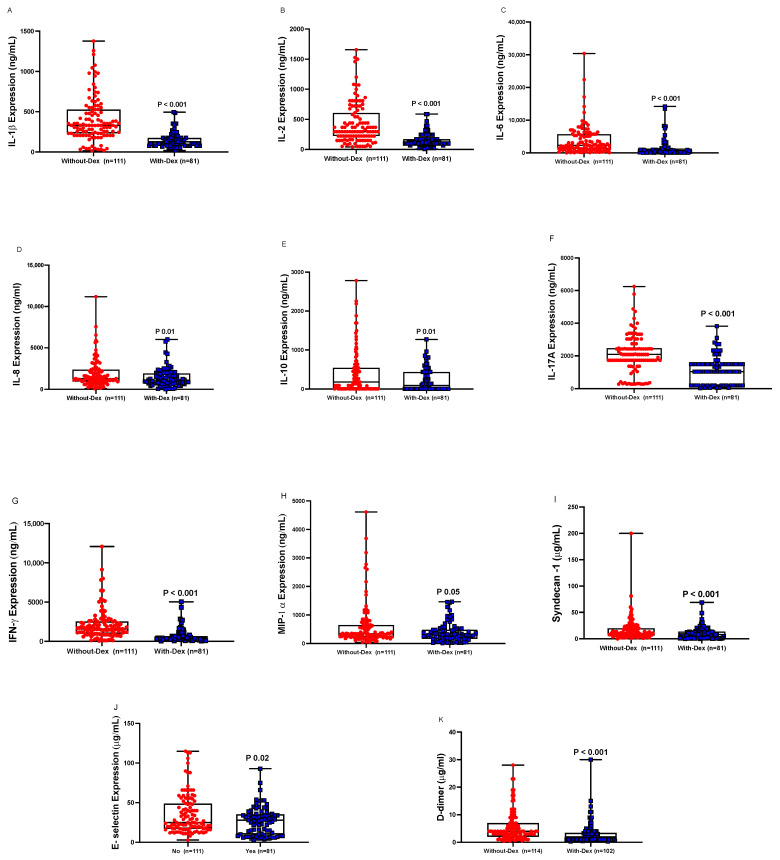
Plasma levels of inflammatory mediators and endothelial injury markers (**A**: IL-1β, **B**: IL-2, **C**: IL-6, **D**: IL-8, **E**: IL-10, **F**: IL-17A, **G**: IFN-γ, **H**: MIP-1α, **I**: E-selectin, **J:** Syndecan-1, **K**: D-dimer) in critically ill COVID-19 patients not receiving treatment (Without Dex), *n* = 110, versus those receiving Dex treatment (With-Dex), *n* = 81, missing: (*n* = 25). Differences analyzed using the Mann–Whitney test.

**Figure 2 biomedicines-11-00164-f002:**
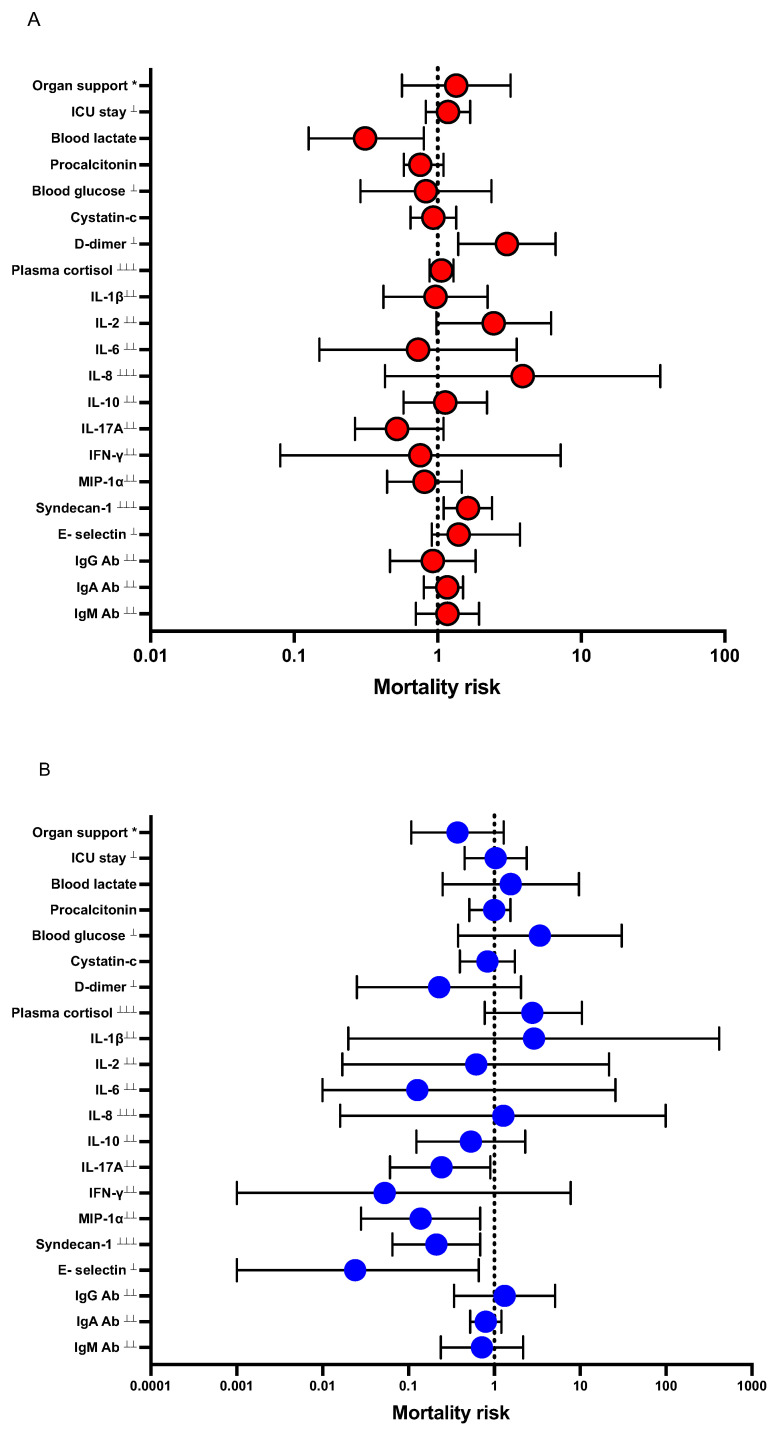
Forest plot showing adjusted odds ratios for effects between listed variables and Dex treatment on 30-day mortality in patients admitted to the ICU with severe COVID-19. Data are presented as odds ratios (OR) and 95% confidence intervals (CI). * Organ support is a composite variable of mechanical ventilation and/or CRRT and/or vasopressors. (**A**), logistic regression model with the variable’s effect on mortality risk as outcome. Model is adjusted for confounders: Dex, comorbidities and SAPS3. (**B**), logistic regression model with the variable’s interaction with Dex treatment on mortality risk as outcome. Model is adjusted for confounders: comorbidities and SAPS3. ^⊥^ variables divided by 10, ^⊥⊥^ variables divided by 100, ^⊥⊥⊥^ variables divided by 1000.

**Table 1 biomedicines-11-00164-t001:** Characteristics of study participants (*n* = 216, by dexamethasone (Dex) administration, no Dex treatment (Without-Dex) versus Dex treatment (With-Dex), presented as *n* (%) or mean ± standard deviation.

		Without-Dex *n* = 114 (52%)	With-Dex *n* = 102 (47%)	*p* Value ^e^
Cohort characteristics				
Age (years)		62 ± 15	66 ± 12	0.09
BMI ^a^ (kg/m^2^)		29 ± 5.8	30 ± 7.1	0.47
Sex	Male	88 (77)	72 (71)	0.38
COVID-19 day ^b^ on ICU ^c^ admission		11 ± 4.4	12 ± 4.1	0.10
SAPS3 ^d^		53 ± 10	55 ± 8	0.43
Comorbidities	Hypertension	61 (53)	77 (76)	**0.001**
	Cardiovascular disease	5 (4)	17 (17)	**0.003**
	Pulmonary disease	30 (26)	20 (20)	0.24
	Diabetes mellitus	32 (28)	35 (34)	0.32
	Chronic kidney disease (stage 3–5)	40 (35)	42 (42)	0.30

^a^ BMI: body mass index. ^b^ COVID-19 day: number of days between symptom onset of Coronavirus disease 2019 (COVID-19) and intensive care admission. ^c^ ICU: intensive care unit ^d^ SAPS-3: Simplified acute physiology score 3. ^e^ *p* value was calculated using Fischer’s exact test or the Mann–Whitney test when appropriate, *p* < 0.05 is significant (bold).

**Table 2 biomedicines-11-00164-t002:** Disease severity and laboratory parameters measured after ICU admission of study participants (*n* = 216), reported for Without-Dex versus With-Dex. Values are shown as *n* (%), mean ± SD, or median ± interquartile range, as appropriate.

	Without-Dex (*n* = 114)	With-Dex (*n* = 102)	*p* Value ^h^
Disease severity, *n* (%)			
Vasopressors	64 (56)	35 (34)	**0.001**
CRRT ^a^	15 (13)	5 (5)	**0.04**
Mechanical ventilation	65 (57)	43 (42)	**0.04**
Hospital-acquired infections	56 (55)	50 (44)	0.10
Length of ICU stay (days)	11 ± 10	8.7 ± 7.0	**0.01**
Mortality (30 days) ^b^	30 (26)	21 (21)	0.32
Laboratory parameters on admission (mean ± SD)			
PaO_2_/FiO_2_ ratio ^c^	20 ± 8.0	20 ± 19	0.61
Blood lactate (mmol/L)	1.5 ± 1.1	1.2 ± 0.5	**0.01**
hsCRP ^d^ (mg/L)	172 ± 82	154 ± 82	0.08
Pro-calcitonin (ng/mL)	1.3 ± 1.7	2.2 ± 1.8	**0.001**
Blood sodium (mmol/L)	136 ± 4.3	137 ± 3.2	0.13
Blood potassium (mmol/L)	3.8 ± 0.4	3.8 ± 0.5	0.39
Blood glucose (mmol/L)	9.0 ± 3.3	10 ± 3.2	**0.004**
Blood creatinine (mg/dL)	100 ± 98	108 ± 125	0.60
Laboratory parameters measured on seven days after ICU admission median (interquartile range)			
Cystatin-C ^e^ (mg/L)	1.1 (0.6–5.2)	1.2 (0.6–6.1)	**0.004**
Plasma cortisol ^f^ (ng/mL)	6.9 (0.05–189)	1.6 (0.01–23)	**0.005**
Plasma ferritin (µg/L)	2255 (107–36,018)	1572 (120–3866)	0.94
Fibrinogen D-dimer (µg/mL)	4.0 (0.5–23)	1.4 (0.3–30)	**0.001**
COVID-19 antibodies ^g^ (*n* = 190)			
IgG (mg/L)	3.5 (0.007–23)	1.9 (0.001–28)	0.07
IgA (mg/L)	0.45 (0.003–24)	0.33 (0.001–7)	0.13
IgM (mg/L)	3.4 (0.006–47)	1.3 (0.00–25)	**0.001**

^a^ CRRT: continuous renal replacement therapy. ^b^ Mortality (30 days): patients who died within 30 days of ICU admission. ^c^ PaO_2_/FiO_2_ ratio: Partial pressure of oxygen in the arterial blood (PaO_2_) gas divided by the fraction of inspired oxygen (FiO_2_). ^d^ hsCRP: high-sensitivity C-reactive protein. ^e^ Cystatin-C: marker of kidney function/glomerular filtration rate. ^f^ Plasma cortisol: cortisol levels in early morning blood sample measured on day seven after ICU admission. ^g^ COVID-19 antibodies: plasma concentration of IgG, IgA, IgM antibodies to spike protein s1, measured on days 10–15 after ICU admission. ^h^ *p* value calculated using the Mann–Whitney test. Bold: *p* < 0.05.

**Table 3 biomedicines-11-00164-t003:** Logistic regression-derived ORs and 95% CIs for the association between Dex and mortality at 30 days in COVID-19 patients admitted to intensive care, (**A**) all patients, (**B**) sub-analysis among With-Dex patients only.

**(A) All Patients (*n* = 216) Admitted to Intensive Care (With-Dex, *n* = 110 vs. Without-Dex, *n* = 114), and Mortality at 30 Days** **.**		
	**Crude**	**Adjusted ^c^**
	OR (95% CI)	OR (95% CI)
With-Dex	0.7 (0.4–1.4)	0.7 (0.3–1.3)
Comorbidities ^a^		0.5 (0.3–1.1)
SAPS3 ^b^		1.1 (0.9–1.1)
**(B) Sub-analysis among COVID-19 patients With-Dex (Early-Dex received Dex treatment within < 14 days of COVID-19 and Late-Dex received ≥ 14 days) vs. Without-Dex and mortality at 30 days.**		
Early-Dex (*n* = 83)	Adjusted ^b^	
With-Dex	**0.4 (0.2–0.8)**	
Comorbidities ^a^	0.6 (0.2–1.3)	
SAPS	1.1 (0.9–1.1)	
Late-Dex (*n* = 19)	Adjusted ^c^	
With-Dex	2.2 (0.4–12)	
Comorbidities	0.4 (0.1–2.5)	
SAPS	1.1 (0.9–1.2)	

^a^ History of hypertension, ischemic heart disease, and/or heart failure. ^b,c^ Adjusted for confounders (comorbidities, SAPS3). Bold: *p* < 0.05.

## Data Availability

Not applicable.

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
