# Peer review of "Immuno-Modulatory Effects of Dexamethasone in Severe COVID-19—A Swedish Cohort Study"

_biomedicines, 2023, doi:10.3390/biomedicines11010164_

Round 1
Reviewer 1 Report
Dr Asif presents a manuscript focusing on the mechanisms underlying the protective effects of dexamethasone during COVID infection.
The paper is well written, divided into appropriate sections, providing a fair inside into the current state of the knowledge as well as fairly discussing the limitations and prospectives. Used English is fine, minor errors were found.
After carefully reviewing the whole draft I have two, very major comments.
The first one is the critical thing - the novelty of the paper is very low. Since the role of dexamethasone in COVID is well known - the world already published over 100 papers in journals like Lancet, Nature Imm, and JAMA. The immune action of this drug is known for over 60 years - the plethora of papers shows graphs like in Figure 1. Moreover, the immuno action of the drug during covid has also been shown and discussed well, please refer to: 10.3390/ijms22136764 ; 10.1038/s41577-020-00421-x; 10.1038/s41577-021-00669-x.
Since the aim of publishing in science is to bring new insights into biology, nature, etc, here there is no new knowledge added to the current state-of-the-art.
The second thing is the data presentation. The manuscript must be redone in terms of statistical analysis. There are substantial errors during analysis. In papers that are prospective observational investigations, data extraction and its proper processing are crucial. Here are the basic errors like: showing non-Gaussian distribution as mean plus minus SD/SEM like for Plasma ferritin; not clearly stated cofounders, lack of algorithm implementations for predictive analysis.
To sum up: this is a very solid paper, if I would reviewing it 2 years ago I would suggest statistical redo and immediate publishing. However, COVID publishing is very time sensitive, and right now, it is too late.
All the best.
Author Response
Answers.
Thank you, for your valuable comments on our paper. The authors agree that a lot has already been published on this topic and the immunological action of this drug is not new. However, we believe that our data adds on insight given its comprehensive coverage of effects presented in the same material enabling assessment of effect sizes. We also present findings that stress the significance of appropriate timings of dexamethasone administration. Furthermore, it also demonstrates the important cytokines and endothelial injury markers which could be responsible for dexamethasone’s role on mortality.
- We have changed the statistical data presentation of ferritin, cystatin-c, D-dimer and COVID-19 antibodies to median and IQR, please see Table-2, on page 4.
- Confounders are mentioned in methods line 161, however after your valuable suggestion we have added additional information on confounders in the results, please see line 231 figure legend line 241, 246, 257.
- We used logistic regression models to study adjusted effects and effect sizes for dexamethasone treatment and outcome. Although the same statistic models are used for predictive analysis this was not an aim of the study.
Reviewer 2 Report
1. Some of the English written need to be carefully revised, such as: Abstract: Patients with dexamethasone treatment vs. “those without”, had higher blood glucose but lower blood lactate, plasma cortisol, IgA, IgM, IgG, D-dimer, cytokines, Syndecan-1 and E-selectin and received less organ support – please revise this sentence.
2. In the abstract: “In a cohort of critically ill COVID-19 pa-tients, early administration of dexamethasone was associated with widespread effects on immune and physiologic responses, some of which could mediate effects on mortality.” I do not understand this sentence, is this a conclusion? Again, please revised this sentence.
3. Please revise the sentence: “patients with dexamethasone treatment and those without” throughout the manuscript. It should be written as: “Patients with and without dexamethasone treatment”.
4. This showed that dexamethasone treatment in patients with severe COVID-19 is associated with widespread effects on the innate and adaptive immune, endocrine, metabolic and coagulation systems and as well as on the endothelial glycocalyx, vascular permeability and organ failure. I think that this is the expected results regarding the Dexamethasone treatment, but not regarding the COVID19 illness.
Other comments are:
1. The biochemical test results obtained for the patients treated with Dex are predictable.
2. the Dex treatment, 14 days within COVID19 infection, does this time point have a special significance? If the patient is not seriously ill, will it also speed up the time of virus clearance?
3. The whole study is about the effect of Dex treatment, and the experimental design only uses 14 days as the cut-off point, so the experimental design is a bit simple. Or is it that the same result will be observed in patients with similar respiratory tracts who develop severe disease later on, as long as they are treated with Dex?
Author Response
Review 2.
- Some of the English written need to be carefully revised, such as: Abstract: Patients with dexamethasone treatment vs.“those without”, had higher blood glucose but lower blood lactate, plasma cortisol, IgA, IgM, IgG, D-dimer, cytokines, Syndecan-1 and E-selectin and received less organ support – please revise this sentence.
- Thank you for your valuable suggestions, we have revised the manuscript accordingly. Please see lines 23-24.
- In the abstract: “In a cohort of critically ill COVID-19 patients, early administration of dexamethasone was associated with widespread effects on immune and physiologic responses, some of which could mediate effects on mortality.” I do not understand this sentence, is this a conclusion? Again, please revised this sentence.
- The authors are grateful for this pointer and this sentence has been revised. Lines 30-31.
- Please revise the sentence: “patients with dexamethasone treatment and those without” throughout the manuscript. It should be written as: “Patients with and without dexamethasone treatment”.
- We have revised the whole manuscript accordingly and the patients are grouped into With-Dex and Without- Dex groups, thank you.
- This showed that dexamethasone treatment in patients with severe COVID-19 is associated with widespread effects on the innate and adaptive immune, endocrine, metabolic and coagulation systems and as well as on the endothelial glycocalyx, vascular permeability and organ failure. I think that this is the expected results regarding the Dexamethasone treatment, but not regarding the COVID19 illness.
- Thank you for the suggestion, we agree that the novelty lies in the context of COVID-19 and have revised the first sentence in the Strengths and limitations line 326, slightly but importantly to emphasize this.
Other comments are:
- The biochemical test results obtained for the patients treated with Dex are predictable.
- Yes, we agree, however we feel it is important to present this data for the readers before presenting the adjusted analyses.
- The Dex treatment, 14 days within COVID19 infection, does this time point have a special significance? If the patient is not seriously ill, will it also speed up the time of virus clearance?
- Thank you for this pointer, the 14 day time point was arbitrarily chosen. We unfortunately have not investigated viral clearance in this study.
- The whole study is about the effect of Dex treatment, and the experimental design only uses 14 days as the cut-off point, so the experimental design is a bit simple. Or is it that the same result will be observed in patients with similar respiratory tracts who develop severe disease later on, as long as they are treated with Dex?
- We agree that the choice of 14 days cut-off is arbitrary. This cut-off was chosen since our studies on COVID changes on lung CT show severe ARDS by this timepoint (PMID: 34348797). Grouping patients into early ARDS and those with fully developed ARDS was important in studying the timing of treatment
Reviewer 3 Report
Asif et al studied the effect of dexamethasone in a Swedish cohort for its efficacy as well s safety in Covid-19. They measured inflammatory cytokines before and after the treatment to see whether dexamethasone reduce these cytokines and have an effect on reducing mortalities. Here are the concerns thay should be corrected before acceptance of the manuscript for publication.
1.In abstract, the value for mortality risk (2.2 (0.4-12) of those who received dexamethasone later should be given. Does mortality risk has any unit, like probability or whether it could be mentioned in text (not in table) as percentage like 4% vs 22% etc for easy understanding to the reader.
2.In methods, 2.1 patients, ICU admission dates, 14th march, but year is missing. It should be given.
3.For readers, It would be better to compress dexamethasone user, nonusers and later users, for example Dex-U, Dex-Nu and Dex-Lu (later user), Dex-early etc.
4.In Table 1 heading, it would be better represented as N=114 (52%) instead of N % 114 (52)
7.In table 1, Dex treated patients have higher CVD and hypertension that is baseline characteristics of the patients’ classifications. This is not the effect of the Dex treatment. This should be written clearly in results.
8. D-dimer differences are not in the figure 1
9. In figure 1, Type 1 interferon gamma that induces host immune response against SARS-CoV2 is decreased in Dex treated group, yet survival is better in Dex treatment. This should be explained
10. In Table 3b heading, what is "Dexamethasone treatment"? Is it morbidities? if so, better to write morbidities only and delete Dex treatment as it may be redundant and difficult to understand the results.
10. The differences in comorbidities for before and after 14days treatment are not very high (Table 3b). Some of the statement about dex mediated reduced morbidities (line 262, 362) in discussion should be softened.
Author Response
Review 3.
Asif et al studied the effect of dexamethasone in a Swedish cohort for its efficacy as well as safety in Covid-19. They measured inflammatory cytokines before and after the treatment to see whether dexamethasone reduce these cytokines and have an effect on reducing mortalities. Here are the concerns thay should be corrected before acceptance of the manuscript for publication.
1.In abstract, the value for mortality risk (2.2 (0.4-12) of those who received dexamethasone later should be given. Does mortality risk has any unit, like probability or whether it could be mentioned in text (not in table) as percentage like 4% vs 22% etc for easy understanding to the reader.
- Thank you for the suggestion, it has been updated, please see lines 28-30.
2.In methods, 2.1 patients, ICU admission dates, 14th march, but year is missing. It should be given.
- Thank you for the pointer, the information is updated. Line 88-89.
3.For readers, It would be better to compress dexamethasone user, nonusers and later users, for example Dex-U, Dex-Nu and Dex-Lu (later user), Dex-early etc.
- Thank you for this suggestion, we have changed the manuscript with abbreviations as suggested. However, since the patients were not users but rather were treated with dexamethasone we chose to call the groups With-Dex, Without-Dex, Early-Dex, Late-Dex.
4.In Table 1 heading, it would be better represented as N=114 (52%) instead of N % 114 (52)
Thank you, this has now been corrected.
7.In table 1, Dex treated patients have higher CVD and hypertension that is baseline characteristics of the patients’ classifications. This is not the effect of the Dex treatment. This should be written clearly in results.
- Thank you, this is now mentioned, please see line 186.
- D-dimer differences are not in the figure 1
- Thank you for the pointer, this has now been updated.
- In figure 1, Type 1 interferon gamma that induces host immune response against SARS-CoV2 is decreased in Dex treated group, yet survival is better in Dex treatment. This should be explained
- Although Interferon-gamma was lower in steroid treated patients this central antiviral mediator did not affect mortalitywhen adjusted for dexamethasone treatment. We don’t report the role of Interferon-gamma on crude mortality
- In Table 3b heading, what is "Dexamethasone treatment"? Is it morbidities? if so, better to write morbidities only and delete Dex treatment as it may be redundant and difficult to understand the results.
- Thank you for this suggestion, we have revised the legends of Table 3a and 3b for clarity.
- The differences in comorbidities for before and after 14days treatment are not very high (Table 3b). Some of the statement about dex mediated reduced morbidities (line 262, 362) in discussion should be softened.
- Thank you we have modified these sentences.
Round 2
Reviewer 1 Report
The Authors in a proper way corrected the form of statistical analysis presentation as well as fixed minor things that were missing in the previous version.
My work as a Reviewer is done here - I still have a feeling that the level of novelty is very low, but the Editor will decide whether it is an obstacle or not to paper be published.
All the best.
Author Response
We thank the reviewer for the comments. Regarding the issue of novelty, we respectfully disagree. This manuscript investigates the effects of dexamethasone of the innate and adaptive immune, endocrine, metabolic and coagulation systems and as well as on the endothelial glycocalyx, vascular permeability and organ failure in critically ill COVID patients. This broad description of biologic effects in one manuscript in a single cohort is novel. Even more importantly we compare the biologic effects of all these pathways enabling assessment of the importance of these on mortality. Given these strengths we feel that this is an important report in the field.